# Explicit Information Placement on Latent Variables using Auxiliary Generative Modelling Task

## Abstract

Deep latent variable models, such as variational autoencoders, have been successfully used to disentangle factors of variation in image datasets. The structure of the representations learned by such models is usually observed after training and iteratively refined by tuning the network architecture and loss function. Here we propose a method that can explicitly place information into a specific subset of the latent variables. We demonstrate the use of the method in a task of disentangling global structure from local features in images. One subset of the latent variables is encouraged to represent local features through an auxiliary modelling task. In this auxiliary task, the global structure of an image is destroyed by dividing it into pixel patches which are then randomly shuffled. The full set of latent variables is trained to model the original data, obliging the remainder of the latent representation to model the global structure. We demonstrate that this approach successfully disentangles the latent variables for global structure from local structure by observing the generative samples of SVHN and CIFAR10. We also clustering the disentangled global structure of SVHN and found that the emerging clusters represent meaningful groups of global structures – including digit identities and the number of digits presence. Finally, we discuss the problem of evaluating the clustering accuracy when ground truth categories are not expressive enough.

## 1 Introduction

Prior knowledge can be imposed on a machine learning algorithm in many ways. One of the most important methods is through the construction of the neural network. A notable example is the use of convolutional neural networks (CNN), where CNNs have shown to improve performance of deep neural networks in many image datasets due to its ability to impose priors about spatiality of the data. The choice of loss function can also be seen as an imposition of prior knowledge, e.g. the square distance error is more suited for a continuous dataset than a discrete dataset.

In this work, we propose a method that can impose prior knowledge onto the representation of latent variable models. In contrast to previous methods, the imposition of prior knowledge is done through the training of a model rather than the construction of a network or its loss function. This method aims to improve the ability of representation learning algorithms with more explicit control over which information is represented at each latent variable.

Many disentangled representation learning methods, such as Variational Auto-Encoder (VAE), implicitly assign the meaning of each latent dimension after training. Since, the representation learned by VAEs is only enforced through the objective function encouraging factorised latent variables. Each latent dimension is only created through the optimisation dynamics which is unknown to the model designer. For example, in the case of the MNIST digit dataset, we have a prior knowledge that the dataset composes of digits with different orientations and thicknesses of strokes. However, it is not possible to explicitly assign the first dimension of the latent variable to represent the orientation while forcing the second dimension to represent the thickness. We can only observe the information placement after a training. If the information can be placed explicitly in the model then different types of prior knowledge can be imposed and, thus improving the model designing process.

To this end, we show that explicit information placement can be made possible through the training setup which we call an auxiliary generative modelling task. We demonstrate the ability of this method through the task of explicitly placing global and local information onto different subsets of latent variables. By doing so, global and local information can be disentangled. We contend that disentangling this type of information is appropriate because it is important prior knowledge which is universally applicable to natural images. For example, a perceptual module of a robot would benefit from learning a separation between the texture and identity of an object. We also propose a suite of experiments that aim to demonstrate that global and local information can be explicitly placed on subsets of the latent variables with our method.

## 1.1 Disentangling local and global structure using VAE

It is well known that a deep convolutional neural network trained on an image recognition task will learn a hierarchy of features, where the lower convolutional layers correspond to local structure and the deeper layers near the classifier neurons respond to higher level features (Gatys et al., 2015; Olah et al., 2018). This disentanglement emerges from the network architecture and the training objective. As the information is pushed through multiple layers of convolutional neurons, it is lossy compressed down to only a class identity in the final output layer.

In a latent variable model such as an autoencoder, however, information is processed differently. In contrast to a recognition model, a latent variable model cannot throw away most of the information in its input layer. Rather, it must reorganise this information into a compact representation which can be used to reconstruct the original data. One way to control the structure of the representation, for example to promote disentanglement, is to adjust the objective functions which control the level of compression and redundancy in the latent variables (Higgins et al., 2017; Chen et al., 2018; Kim & Mnih, 2018; Esmaeili et al., 2018; Dilokthanakul et al., 2016).

Latent variable models (LVMs) assume that the data, $x$, is generated from a latent variable, $z$, and a generative process, $p(x|z)$. One way to fit the LVM to the data is to restrict the generative process to a set of parameterised distributions $p_\theta(x|z)$ and search for parameters $\theta$ that maximise the log-likelihood of the data, $E_{p(x)}[\log p(x|\theta)]$. A Variational Autoencoder (VAE) (Kingma & Welling, 2014; Rezende et al., 2014) is an LVM that parameterises its generative process $p_\theta(x|z)$ and its variational posterior $q_\phi(z|x)$ with deep neural networks, $\theta, \phi$. The parameters of the neural networks are trained with stochastic gradient descent to maximise the variational free-energy objective. The variational free-energy objective is a lower bound to the log-likelihood of the data which can be written as

$$\mathcal{L} = E_{p(x)}[E_{q_\phi(z|x)}[\log p_\theta(x|z)] - KL(q_\phi(z|x)||p(z))]. \tag{1}$$

The VAE is one of the most successful latent variable models, and it has been shown to be effective at modelling image datasets. It has also been shown to be capable of creating latent structure that is compositional and interpretable (Higgins et al., 2017; Kim & Mnih, 2018). These desirable properties are thought to be the result of the free-energy objective, Eq.1, which also has the effect of encouraging the latent variables to be factorised. Eq. 1 consists of two terms known as the reconstruction cost and the KL cost. Following Hoffman and Johnson (Hoffman & Johnson, 2016) and Chen et al. (Chen et al., 2018), the KL term can be rewritten as

$$E_{p(x)}[KL(q_\phi(z|x)||p(z))] = KL(q(z,x)||q(z)p(x)) + KL(q(z)||\prod_j q(z_j))$$

$$+ \sum_j KL(q(z_j)||p(z_j)). \tag{2}$$

We can see from the second term in Eq.2 that the KL cost tries to reduce the total correlation of the latent variables, thus promoting factorised representation.

However, in a complex dataset where images have a variety of textures as well as different global contexts, VAEs struggle to disentangle local structure from global structure and often focus their representations on the local structure which is easier to model. For example, Dilokthanakul et al. (2016) had shown in an SVHN experiment that multi-modal versions of VAEs tend to cluster local structures into the same mode while ignoring the global context, i.e. disentanglement based on the

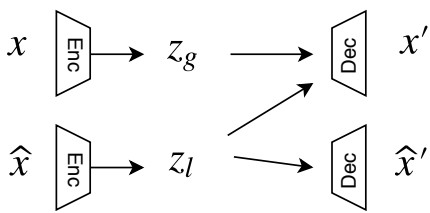

(a) VAE with auxiliary modelling task.



(b) data points, shuffled version and their reconstruction.

Figure 1: VAE consists of an encoder (Enc) and an decoder (Dec). As shown in (a), auxiliary modelling task is an additional VAE with a shared latent variable. The auxiliary VAE performs modelling task on a transformed version of the data. In (b), we show original SVHN data and their shuffled version (Top) and the reconstructions from the VAE (Bottom).

background colours of the image. Importantly, it is not clear whether the factorised representation objective alone would be enough to disentangle this kind of multi-level structure.

Imposing hierarchical structure onto the VAE latent variables, where lower-level latent variables are generated from higher-level latent variables, might be able to disentangle multi-level structure. However, Zhao et al. (2017) argued that hierarchical VAE failed to use the higher latent variables and avoided this problem by only using one level of latent structure (flat latent structure) and carefully creating a neural network architecture that has different levels of computation for each latent dimension. This method successfully disentangles local structure from global structure, but at the cost of needing to carefully tune the architecture, which is the process that is poorly understood. Similarly, Chen et al. (2016) and Gulrajani et al. (2016) showed that information placement can be controlled through the capacity of the decoder. They used the pixelCNN decoder with varying sizes of receptive fields to control how much information the latent variables need to model. However, their methods cannot represent local structures with latent variables, as the local structures will be captured via the pixelCNN, which is an autoregressive model.

## 2 METHODS

### 2.1 EXPLICIT PLACEMENT OF INFORMATION THROUGH AUXILIARY GENERATIVE MODELLING

In this section, we outline our method, which allows the explicit placement of information in each latent variable. Let's denote the original data as a stochastic variable $x$ and a modified version of the data $\hat{x}$. $\hat{x}$ is created from a transformation of $x$ in such a way that the information of interest is destroyed. This transformation has to be crafted for different types of information, e.g. global information can be destroyed by shuffling patches of the images, or colour information is destroyed by conversion to grey scale.

Next, we create two sets of latent variables, $z_i$ and $z_r$. We aim to model the information of interest with $z_i$ and the remainder with $z_r$. We assume a generative processes as follows.

$$z_i, z_r \sim p(z_i, z_r) \tag{3}$$
$$x \sim p_\theta(x|z_i, z_r) \tag{4}$$
$$\hat{x} \sim p_{\hat{\theta}}(\hat{x}|z_r). \tag{5}$$

The process of generating $x$ is the usual generative assumption in VAE. In addition to $x$, $\hat{x}$ is assumed to be generated from $z_r$ which is shared with $x$. By also learning the generative model $p_{\hat{\theta}}(\hat{x})$, we can control the representation of $z_r$ by controlling the information contained in $\hat{x}$. In this case, $\hat{x}$ does not hold details about the information of interest and, therefore, forces $z_r$ to model the leftover information after transformation of $x$. Due to the compression objective of VAE, the remaining variable $z_i$ is pressured to represent the information of interest in order to reduce redundancy in the latent variables. This is what we refer to as an auxiliary modelling task, which explicitly places information of interest in a subset of the latent variables.

## 2.2 Definition of global and local structure of images

In this work, we define notions of *global* and *local information* as follow:

- Local information encapsulates the correlations between nearby pixels in an image.
- Global information encapsulates the correlations between pixels that are further away from each other.

With these definitions, we construct a transformation procedure $\hat{x} = g(x)$ that would destroy the information of interest, in this case the global information, from $x$. Therefore, $z_i$ and $z_r$ are expected to represent the global information and the local information respectively. From now on, we will write $z_g$ for the latent variable modelling the global information and $z_l$ for the latent variable modelling local information.

We propose to destroy the global information by randomly shuffling patches of pixels in the image. The shuffling transformation $g(\cdot)$ is done by first dividing an image into $m$ patches of $n \times n$ pixels. Each patch is assigned an index number associated to it. The indexes are randomly shuffled and then the patches are rearranged. This procedure has two effects: (i) the local correlations between pixels inside a patch are preserved and (ii) the long range correlations between pixels are destroyed through random rearrangement.

For example, in the SVHN dataset, we expect $z_l$ to model the colour scheme of the image because the colour correlations will be preserved regardless of the shuffling process. If one pixel is blue it is more likely that nearby pixels are also blue. We also expect $z_g$ to represent the remaining correlations which should include the identity of the digits and their global styles. In the next section, experiments are designed to measure the extent to which these expectations are met and, therefore how much the information can be explicitly placed on each latent variable using our method.

## 3 Experiments

For our experiments, we used the following datasets:

1. SVHN (Netzer et al., 2011) consists of 604388 training and 26032 test 32x32x3 RGB images of street numbers obtained from Google street view.
2. CIFAR10 (Krizhevsky, 2009) consists of 60000 32x32 colour images in 10 classes, with 6000 images per class including images of trucks, cats, horses, etc.

The aim of the experiments are to test whether the local and global information can be successfully disentangled which would provide an evidence that the proposed method can perform explicit information placement.

### 3.1 Experiment 1: Disentanglement in a simple VAE

In Experiment 1, we trained a simple VAE with our auxiliary generative modelling task. This VAE models the data distribution of $x$ and $\hat{x}$ with two sets of latent variables $z_g$ and $z_l$; $p(x, \hat{x}) = \int_{z_g, z_l} p(x, \hat{x}|z_g, z_l)p(z_g, z_l)dz_g dz_l$. The variational posterior $q_\phi(z_g, z_l|x, \hat{x})$ is modelled as a deep convolutional neural network $\phi$. The encoding process takes a pair of $x$ and $\hat{x}$ as an input through the variational posterior (encoder) $q_\phi(\cdot)$ and output diagonal Gaussian parameters $(\mu_{z_g}, \sigma_{z_g})$ and $(\mu_{z_l}, \sigma_{z_l})$ for $q_\phi$. Next, the samples $z_g, z_l \sim q_\phi(z_g, z_l|x, \hat{x})$ are pushed through the decoder networks $p_\theta(x|z_g, z_l)$ and $p_{\hat{\theta}}(\hat{x}|z_l)$ which are modelled as discretized logistic distributions (Kingma et al., 2016). We then optimise $\phi$, $\theta$ and $\hat{\theta}$ with the standard Monte-Carlo estimate of the free-energy objective,

$$\mathcal{L} = \log p_\theta(x|z_l, z_g) + \log p_{\hat{\theta}}(\hat{x}|z_l) + \beta KL\left(q_\phi(z_g, z_l|x, \hat{x})||p(z_l, z_g)\right), \tag{6}$$

where $\beta$ is a hyperparameter adjusting the compression terms which had been shown to help improve disentanglement (Higgins et al., 2017). The prior $p(z_l, z_g)$ is a unit diagonal Gaussian.

The variational posterior $q_\phi(z_g, z_l|x, \hat{x})$ is assumed to be factorised such that $q_\phi(z_g, z_l|x, \hat{x}) = q_{\phi_g}(z_g|x)q_{\phi_l}(z_l|\hat{x})$. This means we can use two separate neural network encoders $\phi_g$ and $\phi_l$ to

encode the latent variables as shown in Fig 1. This choice of factorisation is chosen to prevent global information going through $z_l$ which means $z_l$ can only represent the information in $\hat{x}$. Although this factorisation is useful, it is conceivable that $z_l$ could benefit from amortising information from $x$ by sharing an encoder.

### 3.1.1 EXPERIMENT 1.1: VISUAL INSPECTION OF GENERATIVE SAMPLES

One way to judge the quality of the learnt latent variables is to inspect samples generated from them at different values. We examine the local latent variables $z_l$ by randomly sampling a point of $z_g \sim p(z_g)$ and 100 points of $z_l \sim p(z_l)$. We then generate 100 generative samples by passing these codes through the decoder. We repeat the same process for the global variables but with one point of $z_l$ and 100 points of $z_g$.

As shown in Fig 2, varying $z_l$ results in variation in colours and brightness while $z_g$ dictates the object identity and global style. This shows that the method works as intended. Similar to the normal VAE, our generative samples of CIFAR10 images look relatively blurry and their object identity is hard to identify. However, the disentanglement between colour and global structure can be observed. Several $\beta$ values had been used and it had been observed that the disentanglement between global and local information is robust to different values of $\beta$.

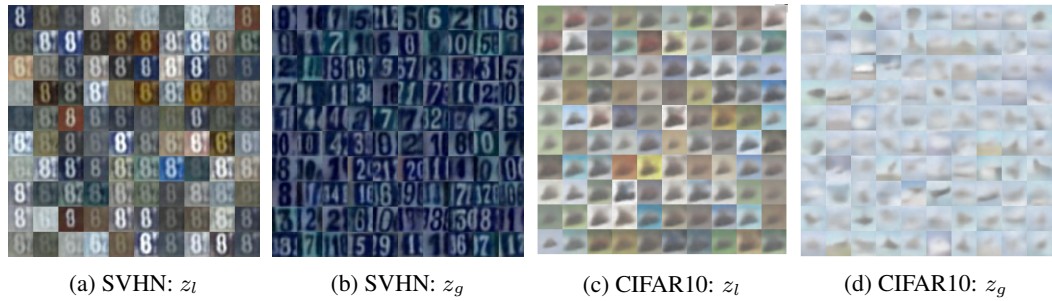

| (a) SVHN: $z_l$ | (b) SVHN: $z_g$ | (c) CIFAR10: $z_l$ | (d) CIFAR10: $z_g$ |

Figure 2: Inspecting local and global latent variables. We show that by varying the local latent variable $z_l$ the background colours and brightness of the generative images change (see (a) and (c)). The global variable $z_g$ dictates the digit identity and style for SVHN and global structure for CIFAR10 while keeping the background colour and brightness unchanging (see (b) and (d)). $\beta = 20$ was used to produce the figures.

### 3.1.2 EXPERIMENT 1.2: QUANTITATIVE INSPECTION OF DISENTANGLEMENT

In order to show more *quantitatively* that the method can explicitly place global information in a subset of the latent variables, an experiment is carried out where an SVHN classifier is trained to inspect the generative samples of the model. The classifier has an accuracy of 95% on the SVHN test set. The SVHN test data is encoded into the latent space with the encoder of the model used in previous experiment with $\beta = 1.0$. Then, three types of samples are generated from the encodings: (i) images generated directly from the encoded latents, (ii) images in which $z_l$ is replaced with a random sample from $\mathcal{N}(0, 1)$ while preserving $z_g$, and (iii) images in which $z_g$ is replaced with a random sample from $\mathcal{N}(0, 1)$ while preserving $z_l$.

The result in Fig. 3 shows that: (i) the direct reconstruction slightly perturbs digit identity, yielding an accuracy of 86%, (ii) changing $z_l$ also slightly perturbs the digit identity, yielding 80% accuracy, (iii) while changing in $z_g$ completely changes the identity of the digits, reducing the accuracy to 11%. The difference between 80% and 11% (ie: chance) in (ii) and (iii) demonstrates quantitatively the disentangling we were aiming for. Increasing $\beta$ results in blurred reconstruction images and results in lower classification accuracy.

### 3.2 EXPERIMENT 2: DISENTANGLEMENT IN MULTI-MODAL VAE

In previous experiments, we have shown that changing $z_l$ only slightly effects the identity of digit while changing $z_g$ significantly altered the digit identity. In this experiment, we would like to inspect

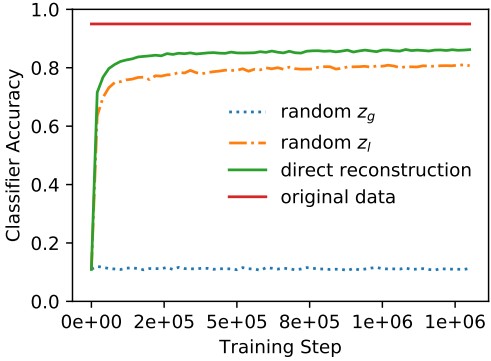

Figure 3: Classifier accuracy of the generated test samples. The classifier has the accuracy of $95\%$ on SVHN test set. The accuracy is reduced to $86\%$ when the test data is auto-encoded through the model. By changing the $z_g$, the identity of the digits are completely changed. However, this is not the case when $z_l$ is changed.

the content of $z_g$ in more details, as well as demonstrating the use of our method in an unsupervised clustering task.

The model in experiment 1 is modified with an additional discrete latent variable, $y$, resulting in the following generative process:

$$z_l \sim p(z_l) \tag{7}$$
$$z_g, y \sim p_\gamma(z_g|y)p(y) \tag{8}$$
$$x \sim p_\theta(x|z_g, z_l) \tag{9}$$
$$\hat{x} \sim p_{\hat{\theta}}(\hat{x}|z_l). \tag{10}$$

The variational posterior is assumed to be factorised as $q_\phi(y, z_g, z_l|x, \hat{x}) = q_{\phi_g}(y, z_g|x)q_{\phi_l}(z_l|\hat{x})$ with diagonal Gaussians as the posteriors of continuous variables $z_g$, $z_l$ and a Gumbel-Softmax (Concrete) distribution (Maddison et al., 2017; Jang et al., 2017) for the class variable $y$ with constant temperature $\tau$. Finally, we optimise the free-energy objective:

$$\begin{aligned}\mathcal{L} = &\log p_{\theta, \hat{\theta}}(x, \hat{x}|z_l, z_g, y) \\ &+ \beta KL(q_{\phi_g, \phi_l}(z_l, z_g|x, \hat{x})||p_\gamma(z_l, z_g|y)) \\ &+ KL(q_{\phi_g}(y|x)||p(y)),\end{aligned} \tag{11}$$

where $\beta$ is the compression pressure on latent variable $z_l$ and $z_g$.

### 3.2.1 EXPERIMENT 2.1: DIGIT CLUSTERING ON THE GLOBAL VARIABLE

We evaluated the model in the digit's identity clustering task. This is a standard evaluation metric for clustering methods. The model is evaluated with the clustering accuracy (ACC) on the test set after training and hyper-parameter tuning on the training set. The training and evaluation are repeated with the same hyper-parameter setting for four runs. ACC calculates the accuracy by assigning the best ground truth label for each cluster and, thus, allows for a larger number of predicted cluster assignments than ground truth labels. $\beta = 40$ was searched from $\beta \in \{1, 10, 20, 30, 40, 50, 60\}$ for the highest digit's identity clustering result.

As shown in Table 1, the method achieves a competitive score to dedicated clustering algorithms when clustering SVHN identities. This supports earlier results that the local structure can be disentangled away with an auxiliary task.

However, there are many ways to cluster SVHN's global image structure which can be much richer than the identity of the middle digits and, as such, this evaluation benchmark cannot be used to fairly compare the ability of clustering algorithms.

Table 1: SVHN Digit Identities Clustering Results

| Model | k | ACC (%) |
|---|---|---|
| VAE + Auxiliary Task (Our) | 30 | 58.2 ($\pm$ 5.6) |
| DEC (Xie et al., 2016) | 10 | 11.9 ($\pm$ 0.4) |
| IMSAT (Hu et al., 2017) | 10 | 57.3 ($\pm$ 3.9) |
| ACOL + GAR + k-means (Kilinc & Uysal, 2018) | 10 | 76.8 ($\pm$ 1.3) |

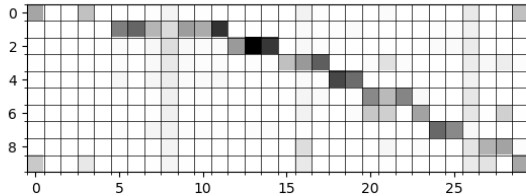

Figure 4: Digit identity clustering assignments on the SVHN test set. The y-axis indicates 10 ground truth labels and the x-axis indicates 30 predicting clusters (re-arranged). The darker the box the more data points assigned to the cluster. Each ground truth identity is represented by approximately three clusters. Interestingly, digit "1" uses 7 clusters which is much more than other digit classes. This is likely to be the results of distractors in images with digit "1" present. There is a clear confusion between digit "5" and "6" which are sharing some of the clusters.

Although $z_g$ is encouraged to contain global structure, it is not given that the latent $y$ will hold the information about the digit class rather than encoding it in $z_g$. Some SVHN data points have 'distracting' digits on the sides of the digit of interest (middle digit). It makes sense that some clusters $y$ could be based on whether or not an image has distracting digits. In fact, these kinds of clusters were frequently observed. Fig 5 shows the clusters that are based on the presence of distracting digits.

This result shows that (i) $z_g$ mainly contains the information about digit information and the number of digits in an image, which are global information, and (ii) it suggests that the way the unsupervised clustering accuracy metrics are usually reported has a flaw which is not discussed in earlier deep clustering works. The architecture is normally tuned to achieve the wanted clustering preference rather than relying on robust information placement. This problem presents itself when the ground truth labels are not expressive enough.

### 3.2.2   EXPERIMENT 2.2: VISUALISING THE GENERATIVE SAMPLES

Next, we inspect the latent variables $z_g$ and $y$ of a chosen architecture. As shown in Fig 6, we found that $z_g$ represents the digit's global style, e.g. orientation and stroke width, while $y$ represents the digit identity and whether or not the digit is darker than the background colour. These clusters are interesting because they represent the *global* colour style, in contrast to the local colour style modelled with $z_l$.

This visualisation further confirm that $z_g$ and $y$ tend to model global information which supports our hypothesis that the method can be used to explicitly place information onto a subset of the latent variables.

## 4   RELATED WORKS

Several specialised clustering systems have attempted to cluster the SVHN dataset according to digit identity (Hu et al., 2017; Kilinc & Uysal, 2018). However, there is no clear justification for why these algorithms should cluster digit identity rather than the background colour or texture. It is also not clear how much architectural tuning is needed to bias the clustering preference of these algorithms.

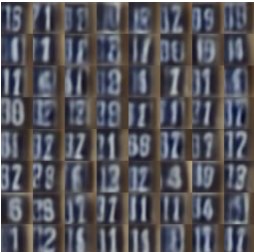 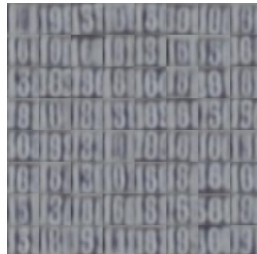

Figure 5: Clustering preference. The figures show generative samples from two clusters $y$ that represent images with two digits (left) and three digits (right). The local code $z_l$ is fixed for each cluster and, therefore, the samples have the same colour scheme. The global codes $z_g$ are randomly sampled which changes the style and identity of the digits.

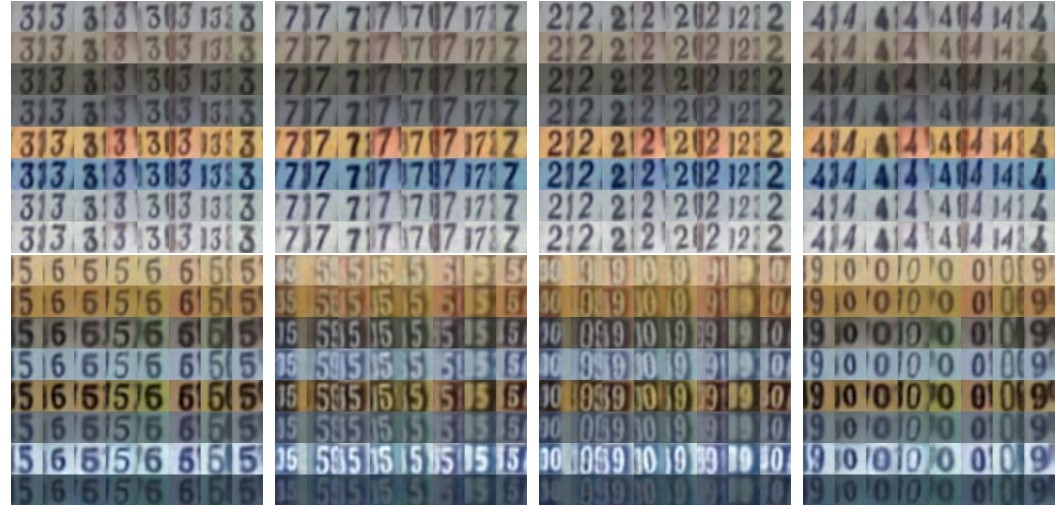

Figure 6: Generative images from different clusters $y$. Each sub-figure shows that each cluster mainly represents an identity of the digits. However, two digit identities might share the same cluster (e.g. "5, 6" and "0, 9"). In addition to the digit identity, the VAE also disentangles images into groups which either have darker digits than the background or lighter digits than the background. Within each sub-figure, we vary the global latent code (column) and local latent code (row) showing clear disentanglement between the background colour and digit style.

Notably, an architecture called TAGGER (Greff et al., 2016) can learn a disentangled representation of background and objects by explicitly imposing an assumption on the network architecture that a representation of an object consists of a linear combination of location masks and texture patterns. However, TAGGER separates objects from background at the pixel level and has not been shown to scale-up to a bigger and more natural dataset which might require separation at the level of the latent space.

The topic of disentangled representations has gained plenty of recent attention from researchers, with much of the effort directed at studying latent variable models such as VAEs. Much of this work tries to dissect the objective of the VAE (Hoffman & Johnson, 2016; Chen et al., 2018; Esmaeili et al., 2018) and proposes modified objectives that emphasise different aspects of the latent structure. However, using objective modifications alone, it is difficult to scale disentanglement to more challenging datasets.

The closely related work by Zhao et al. (2017) tackled the same problem of multi-level structure disentanglement with VAEs. In their work, the disentanglement is achieved through careful architectural design, where more abstract information goes through more computational layers. The information placement is very sensitive to the architectural design and needs an iterative design pro-

cess. Our method achieves similar disentanglement through an alternative route, with more explicit control on what information is to be placed on specific latent variables.

The Attend Infer Repeat (AIR) model (Eslami et al., 2016) is another example of information placement using architectural design. AIR has achieved the disentanglement of object's identity, location and quantity through the use of specialised encoders and decoders. While AIR has not been able to be scaled up to more complex datasets such as multi-digit SVHN, it would be interesting to combined the ability to disentangle global structure of our method with AIR's ability to count and identify objects. This is left for future work.

DC-IGN (Kulkarni et al., 2015) can perform explicit information placement through the training process. However, the method need the knowledge of the feature variations in batches of data (similar to having access to the labels of the data). Similar to our method, Tranforming autoencoders (Hinton et al., 2011) uses transformation of original images as self-supervision signals. However, the goal of the method is to learn the factor corresponding to the transformation. Implicit Autoencoder (Makhzani, 2018) shows that adversarial regularisation can force a latent variable to model only global information as the information has to be as compact as possible. This work shows a possibility of implicitly imposing inductive bias in the algorithmic design.

## 5 CONCLUSIONS

In this paper, we proposed a training method which explicitly controls where the information is to be placed in the latent variables of a VAE. We demonstrated the ability of the method through the task of disentangling local and global information. We showed that the latent variables are successfully disentangled by observing generative samples of the SVHN and CIFAR10 datasets. This experiment also shows that the local structure and global structure of generated images can be controlled by varying the local and global latent codes.

Next, we attempted to quantify the quality of the global latent variable through SVHN clustering. We inspected the clusters of the global variable and observed that it represents global information such as digit identity, number of digits and the global colour style. In addition, we observed that the local colour style is modelled with the local variables as expected. These experiments give substantial evidence that our method can be used to explicitly place global and local information onto the subset of the latent variables.

ACKNOWLEDGMENTS

Acknowledgement will only be visible in unanonymised version.

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
