# OpenReview forum: "Explicit Information Placement on Latent Variables using Auxiliary Generative Modelling Task"
_ICLR.cc/2019/Conference_

### Official Review · AnonReviewer3 · 2018-11-02
**Approach seems limited and the source of improvement is not clear**

**Rating:** 5
**Confidence:** 4

**Review:**

This work proposes an approach for explicitly placing information in a subset of the latent variables. The approach is to construct an auxiliary generative model that takes as input the set of latent variables subtracted from the target subset, which is used to model modified data samples that do not contain the desired information.

Experiments focus on learning global information. The auxiliary model is then given data that have their global information destroyed via random shuffling of image patches.

# Approach seems limited.
 - This approach seems very limited, as there must exist a known transformation that removes the desired information. Apart from global vs. local, can the authors provide more examples of what sort of information this approach can disentangle? (Even for global vs. local, is there a transformation that can remove local information as opposed to global information?)
 - Can this approach learn multiple factors as opposed to just two?
 - What if the desired factors are not clearly disjoint and collectively exhaustive? (e.g. mustache vs. gender on human faces.)

# More ablations or experiments with comparable settings would be desirable.
 - What is the choice of beta in the beta-VAE training objective? Apart from 1.2, this isn't mentioned. My concern here is that beta might be affecting the result more than the proposed training algorithm. Can the proposed approach perform just as well without a modified objective? Ablation studies that show the proposed algorithm can improve upon the baseline in all settings would make this a stronger paper. (e.g. this approach with normal VAE objective, and normal VAE objective without auxiliary task for the clustering experiment.)
 - Why were 30 discrete categories used in the clustering experiment? Is this still comparable to the approaches that use 10, which would correspond to the number of classes?

# Related work.
There are some well-cited works that the authors may have missed. These are ultimately different approaches, but perhaps the authors can obtain some inspiration from these:
- Tranforming autoencoders [1] also apply a transformation to the image, but the goal is to learn the factor corresponding to the transformation, rather than the complement as in this work.
- An opposing approach for explicit information placement with a modified training procedure (where the target information is directly placed in the target subset and can handle multiple factors) is DC-IGN [2]. I believe the DC-IGN approach is more general and can handle a superset of the tasks of this approach, without requiring an auxiliary decoder. Comparing to this approach, I wonder if it would be better to provide samples that exhibit a particular factor, or samples that conceal the factor?

[1] Hinton, Geoffrey E., Alex Krizhevsky, and Sida D. Wang. "Transforming auto-encoders." International Conference on Artificial Neural Networks. Springer, Berlin, Heidelberg, 2011.
[2] Kulkarni, Tejas D., et al. "Deep convolutional inverse graphics network." Advances in neural information processing systems. 2015.

---- Update since rebuttal ----

I thank the authors for clarifying how this work fits in with related works and clarifying the hyperparameters. I maintain my concerns that the experiments are limited and do not showcase the individual benefit of using explicit information placement. More experiments based on different transformations that the authors have mentioned would make this a stronger contribution. The use of beta>1 is fine if it helps alongside the use of this approach, but it would have been better to see the effects of this approach and beta>1 (and other hyperparameters such as k in Table 1) in isolation.

---

> ### Author Response · Authors · 2018-11-19
> **Rebuttal**
>
> We thank reviewer3 for the comprehensive review. We would like to address each of the reviewer's concerns individually. (This will include some discussions above only seen by authors, reviewers, area chairs and program chairs.)
>
> # Is the approach really limited?
>
> It is true that that the transformation that removes the desired information must be known before hand which is the main assumption in the paper.
> Other conceivable transformations are (i) removal of colour information by converting image to grey scale, (ii) removal of orientation information with random rotation and positional shift, (iii) removal of temporal correlation using shuffling in a time-series data and etc. We hope that that this paper could ignite a discussion around what transformation can be created to impose prior knowledge into the model. These prior transformation could be something that we observed in biology, for example, we observe global-local information disentanglement in our perception. Is there other hard-coded disentanglement in biology? This is rather an interesting problem in our opinion.
>
> Can this method learn more factor than just two? It is conceivable that there could be more than one information of interests that get destroyed in a transformation. For example, one latent factor could model middle-range correlations if the transformation remove long-range correlations through shuffling process and short-range correlations get destroyed through a blurring process (e.g. local smoothing transformation). Another two factors could represent long-rang and short-range correlations.
>
> What if the desired factors are not clearly disjoint and collectively exhaustive? This is an interesting question. We do not think that our current approach can disentangle continuous features. In a future work, there could be an auxiliary task method that can create continuous latent variables. We hope that this paper create interesting open problems for future research.
>
> # More ablations or experiments with comparable settings would be desirable.
>
> In our experiments, we found that the disentanglement of global and local information is very robust to different values of beta. In experiment 1.2 we use beta=1.0 which is the same as using the original VAE objective. However, beta does affect the quality of the generative samples (blurriness). For experiment 1.1, different beta produce similar disentanglement results, we use beta=20 to produce the figures as it created nicest looking samples. We uses beta=40 for all clustering experiments which had been searched from beta=\{1, 10, 20, 30, 40, 50, 60\} for the best digit identity clustering results. Thanks to reviewer3, we incorporated this information into the revision.
>
> Regarding the clustering result, we believe that the resulting accuracy number cannot be used to compare the quality of the clustering methods. We observe that the global structure contains more information than just digit identity. It also contains information such as whether or not there are distracting digits in the image. We are not concern with improving upon baseline but rather to confirm that our method can disentangle global-local information and the further analysis have shown that the grouping corresponds to more than just the digit identity. The use of 30 clusters helps us identify the grouping of other types of global information in addition to the digit identity. Therefore, the identity clustering performance does not directly translate into the ability to disentangle local and global variables.
>
> # Related work
>
> We would like to thank reviewer 3 for suggesting the related works that we have missed. These were incorporated in the revision.
>
> As discussed in the comments above (visible only to authors and area chairs), there is an overhead regarding grouping of data into batches in DC-IGN approach. We agree that DC-IGN could potentially perform the same task as our model or more. However, our method can reduce the effort of needing to group the data or use labelled data by instead thinking more about prior knowledge (transformation function) of the entire dataset.
>
> The contributions of this paper are
> (i) Suggest that there is another method of imposing prior knowledge into algorithmic design of the latent variable model. We believe this can be categorised as a self-supervised learning approach (a kind of unsupervised learning) which have not been explored much in the context of the latent variable model.
>
> (ii) Show that it can be used to disentangle global and local information through experiments.

---

### Official Review · AnonReviewer2 · 2018-11-03
**Good paper; a broader analysis beyond global-local disentanglement is desirable.**

**Rating:** 7
**Confidence:** 4

**Review:**

The paper proposes a method to disentangle latent variables for certain factors of interest in an image by considering the original input image and a transformation of the image where information about the factors of interest is removed. The generative process is then modeled by having two latent variables --  the first responsible for generating the transformed image whereas both latent variables are responsible for generating the original input image. This inductive bias naturally enforces that the second latent variable will not model the information which the first needs to reconstruct the transformed image, due to the VAE objective penalizing redundancy in information present in the latents. The paper demonstrates this in one setting where the transformation is random shuffling of image patches, which should remove the global information of the original input image.

The methodology of the paper was concise and easy to follow. The simple inductive bias presented in the paper for disentangling local and global information is very interesting. It is not obvious that shuffling image patches at a particular scale would lead to complete loss of global information, but the paper does show results on SVHN and CIFAR10 for which global information is sufficiently disentangled. The results for digit identity clustering were great for showing the correlation between their learnt global information and label information.

The paper introduced their model as a general purpose strategy for placing desired information in latent variables using auxiliary tasks, but focus was directed to the global vs local line of analysis. While giving examples for what kind of information can be removed, the authors mentioned that color to gray-scale might be one possibility.  It would have been interesting to see this and other possibilities explored in the paper. I feel that the idea deserves a broader analysis beyond just a single choice of disentanglement.

It is mentioned in the paper that having a single inference network for the posterior as opposed to the factorized one is conceivable. I would be curious to see an analysis of how that works out as compared to the separate encoders case.

Overall, the paper has a novel idea which is well motivated and executed in terms of experiments.

---

> ### Author Response · Authors · 2018-11-19
> **Thanks for your review**
>
> We thank reviewer2 for the kind and constructive review.
> We agree that broader analysis beyond global-local disentanglement is desirable and we hope to perform more experiments in a follow up work.

---

### Official Review · AnonReviewer1 · 2018-11-04
**An interesting model without enough experimental justifications**

**Rating:** 6
**Confidence:** 4

**Review:**

This paper proposed a new training framework to disentangle global structures from local structures
based on Variational Autoencoders (VAEs). They first generate a transformed image by shuffling the
patches of the original image to destroy the global structures. The training task forces the model to
reconstruct the original image and shuffled images from different latent variables, thus separating
global long-range structural correlations and local patch-wise correlations .

Instead of adjusting the objective function or model structure, the paper proposed a new and simple
training framework to disentangle the global and local structures, which is novel.

The experiment results are good on SVHN. Some visual inspection experiments on CIFAR10 are performed.
The plot (Figure2 (d)) is very blurry and people cannot really tell local structure from it. The rest experiments
are all based on SVHN, which is too simple.

More experiments based on other types of data sets with clear global structures such as faces or stop signs will
be more convincing.

In the digit dataset, the local and global structures are relatively easy to separate. However, in Table 1, the
performance of VAE+Auxiliary is not better than two of the other methods.

The idea in this paper is novel but experiments do not seem to be enough. More experiments on datasets
with clear global and local structure separations with careful analyses are required to make the paper stronger.

---

> ### Author Response · Authors · 2018-11-19
> **Thanks for your review**
>
> We would like to thank Reivewer 1 for your time and the review.
> We agree that more experiments based on other types of dataset will make the result stronger which we hope to perform in a follow up work. However, we believe that the current results already give substantial evidences that the method successfully disentangle local and global structure.
>
> While the performance of VAE+Auxiliary in digit identity clustering is not higher than two of the other methods, we found that the grouping can corresponds to other global features such as how many digits are in the image and the global-colour style. Therefore, the lower clustering accuracy does not mean that the method poorly disentangle local and global information but rather suggesting that the digit identity clustering is an incomplete evaluation metric for unsupervised clustering. We hope that reviewer 1 see that, in the context of this paper, the experiments have substantially fulfilled their proposed of showing that the method can disentangle global and local information as intended.

---

### Author Response · Authors · 2018-11-27
**Changes made in the revision**

Dear Reviewers and AC,

We added details about hyper-parameters (beta) used in the experiments. We also added a discussion that this parameter does not significantly affect the disentanglement between global and local variables but it does affect the blurriness of the reconstruction and the clustering preference of the clustering model.  We also added discussions about related works suggested by reviewer3 in the related work section.

We want to thank reviewers and AC again for their time. We believe that this work contains an important idea that contributes  towards better representation learning method. The idea, that an inductive bias can be imposed through an auxiliary task. This alternative way of imposing inductive bias opens the possibility of combining the usual way of imposing inductive bias (through network design) with the new way of using an auxiliary task. Explicitness in the imposition of prior knowledge also makes the algorithm design much more transparent. Our experiments confirm that our method can disentangle local- and global structure as we expected.

In the context of this paper, we contend that our experiments strongly support the hypothesis that it can be used to explicitly disentangle local and global structure. This level of explicitness of disentanglement between local- and global variables shows that the method is significantly different and it also can complement the existing methods.  We agree that more experiments can be used to find out whether this method can be generalised into other problems in addition to the local-global disentanglement which we agree will be good additional contributions.

---

### Meta-Review · Area_Chair1 · 2018-12-14
**Interesting idea, but the novelty is not high and the experimental analysis is weak**

**Confidence:** 5
**Recommendation:** Reject

**Metareview:**

While the paper has good quality and clarity and the proposed idea seems interesting, all three reviewers agree that the paper needs more challenging experiments to justify the proposed idea. The authors are not able to include additional experiments (such as these based on different transformations) into their revision to better convince the reviewers. In addition, the AC feels that the technical novelty of the paper is rather minor (some incremental change to VAE). In particular, related to some concerns of Reviewer 3, the AC feels the proposed idea is not too much different than introducing certain kind of side-information for supervision; the main novelty seems to be distorting the data itself somehow to provide these side information (which does not seems to be that novel).